# Multidisciplinary interventions for reducing the avoidable displacement from home of frail older people: a systematic review

Lucas Sempé ![ORCID],[1] Jenny Billings,[2] Peter Lloyd-Sherlock[1]

¹School of International Development, University of East Anglia, Norwich, UK
²Centre for Integrated Care Research, University of Kent, Canterbury, UK

**Correspondence to**
Lucas Sempé;
l.sempe@uea.ac.uk

## ABSTRACT

**Objectives**  To synthesise existing literature on interventions addressing a new concept of avoidable displacement from home for older people with multimorbidity or frailty. The review focused on home-based interventions by any type of multidisciplinary team aimed at reducing avoidable displacement from home to hospital settings. A second objective was to characterise these interventions to inform policy.

**Design**  A systematic search of the main bibliographic databases was conducted to identify studies relating to interventions addressing avoidable displacement from home for older people. Studies focusing on one specific condition or interventions without multidisciplinary teams were excluded. A narrative synthesis of data was conducted, and themes were identified by using an adapted thematic framework analysis approach.

**Results**  The search strategy was performed using the following electronic databases: the American National Library of Medicine and the National Institutes of Health (PubMed), Scopus, Cochrane Library (Central and CDRS), CINAHL, Social Care Online, Web of Science as well as the database of the Latin American and Caribbean Health Sciences Literature. The database search was done in September 2018 and completed in October 2018. Overall 3927 articles were identified and 364 were retained for full text screening. Fifteen studies were included in the narrative review. Four themes were identified and discussed: (1) types of interventions, (2) composition of teams, (3) intervention effectiveness and (4) types of outcomes. Within intervention types, three categories of care types were identified; transitional care, case-management services and hospital at home. Each individual article was assessed in terms of risk of bias following Cochrane Collaboration guidelines.

**Conclusions**  The review identified some potential interventions and relevant topics to be addressed in order to develop effective and sustainable interventions to reduce the avoidable displacement from home of older people. However the review was not able to identify robust impact evidence, either in terms of quantity or quality from the studies presented. As such, the available evidence is not sufficiently robust to inform policy or interventions for reducing avoidable displacement from home. This finding reflects the complexity of these interventions and a lack of systematic data collection.

**PROSPERO registration number**  CRD42018108116.

---

### Strengths and limitations of this study

► This is a timely systematic review of interventions involving integrated care for older people, situated within a policy framework that focuses on a new concept of avoidable displacement from home.

► This review examines interventions addressing a significant public health and social care challenge.

► Evidence in this area is predominately from quasi-experiments and observational studies with a number of studies presenting unclear results and ambiguity regarding the effectiveness of the interventions on hospital and emergency department admissions and readmissions.

---

## INTRODUCTION
### Rationale

Between 2017 and 2050, the global population of people aged 60 years or over is expected to more than double, reaching 2.1 billion people. In 2015, around two-thirds of this population lived in low-income or middle-income countries, rising to 80% of older people world population by 2050.[1] Later life is associated with an increased risk of frailty, poor health and functional limitation.[2] In this paper, frailty refers to a distinctive health state related to the ageing process in which multiple body systems gradually lose their in-built reserves.[3] Population ageing creates challenges for both social services and health systems, including rising numbers of hospitalisations, many of which are potentially avoidable.[4 5] Responses to these challenges include an increased emphasis on interventions to integrate different forms of health and social care provision, working across different care settings.[6] There are several definitions of integrated care: this paper follows Kodner and Kyriacou's[7] who define it as a '... set of techniques and organisational models designed to create connectivity, alignment and co-ordination within and between the

cure and care sectors at the funding, administrative and/or provider levels… for patients with complex problems'. While different degrees of integration are possible, all by definition require the cooperation of multidisciplinary professional teams working together to deliver appropriate care. Multidisciplinary interventions to promote and support integrated care are now widely advocated at many policy levels. It is claimed, however, that the effectiveness of these approaches is inconsistent across care settings and different interventions.[8]

### A new framework: avoidable displacement from home

The concept of avoidable displacement from home (ADH) and an associated policy framework are currently being developed and validated by the authors as part of an Medical Research Council Newton funded project 'Improving the effectiveness and efficiency of Health and social care services for vulnerable Older Brazilians' (IHOB) (2018–2021).[9] This review represents one aspect of the validation process.

Given the wide diversity of integrated care interventions and their complex interactions with other parts of health and social care systems, it is important to establish clear parameters about where they are situated in relation to different care settings and to clarify outcomes of interest. To this end, 'avoidable displacement from home' may provide a helpful framework, by encompassing a number of effects: hospital (re)admissions that are amenable to prevention or which are suited to outpatient care, excessive hospital stays and unnecessary admissions into residential care homes. More broadly, an avoidable displacement from home can be understood as the consequence of challenges to deliver proper care that permits older people to remain in their homes for as long as possible when this is in their best interest. It represents a comprehensive approach to address health and social care as part of a single, integrated system.

Interventions to reduce avoidable displacement from home should aim to incorporate three core values: person-centredness,[10] place-based care[11] and sustainability.[12] In this way, the concept may encourage policy-makers to go beyond an exclusive focus on health system efficiency (such as reducing unnecessary hospitalisations) towards a wider approach that seeks to reconcile health and social service provision with the needs and wishes of both older people and their carers.

### Objective

Our specific objective was to conduct a systematic review to identify interventions addressing displacement from home of older people with multimorbidity or frailty. The review focused on studies with the following characteristics: home-based interventions by any type of multidisciplinary team that aimed at reducing avoidable displacement from home in older people with multimorbidity or frailty. A second objective was to characterise those interventions in order to inform policy.

## METHODS

### Information sources and searches

This is a systematic review of articles indexed in the following electronic databases: the American National Library of Medicine and the National Institutes of Health (PubMed), Scopus, Cochrane Library (Central and CDRS), CINAHL, Social Care Online, Web of Science as well as the database of the Latin American and Caribbean Health Sciences Literature. We also examined the reference lists of the selected studies.

The search was conducted without any language restriction and focused on articles published after the year 2000. The review only included articles published in Portuguese, English and Spanish, because of our familiarity with these languages. The protocol for the related search terms was completed in July 2018 while the full database search was performed in September 2018, and the articles were moved into Mendeley V.1.18. The duplicates have been deleted.

### Search strategy

An initial scoping of the literature was conducted at inception of the study and those findings were used to define the search strategy. In order to take an inclusive approach we included a variety of key search terms, utilising the scope of Medical Subject Headings for each of them. The terms linked three different topics, namely: 'home-based care' interventions, and terms to identify 'older adults' in conjunction with terms used as proxies of preventive outcomes ('Involuntary hospitalisation', 'Avoidable hospitalisation', 'Patient admission' or 'Voluntary Admission') and readmissions ('Patient Readmission', '30 Day Readmission', 'Thirty Day Readmission', 'Hospital Readmissions' or 'Readmissions, Hospital'). The term 'multidisciplinary teams' or related terms were not used in the search to prevent restricting the results. The search syntax and data extraction form are provided in online supplementary appendices 1 and 2.

### Eligibility criteria

Following the review objectives, any type of quantitative study (randomised control trials (RCTs), quasi-experiments and observational data) that fitted our criteria was included in the review process. The eligibility criteria guiding the search enabled the selection of publications specifically focusing on older people (defined as aged over 60 years) who received any kind of home-based intervention by a multidisciplinary team. A multidisciplinary team was defined as a formal team of two or more people from different professions working together to deliver their service. In order to assess any effect on avoidable displacement from home, the studies needed to record at least one of the following outcomes: hospital length of stay, hospital or emergency department admissions or readmissions.

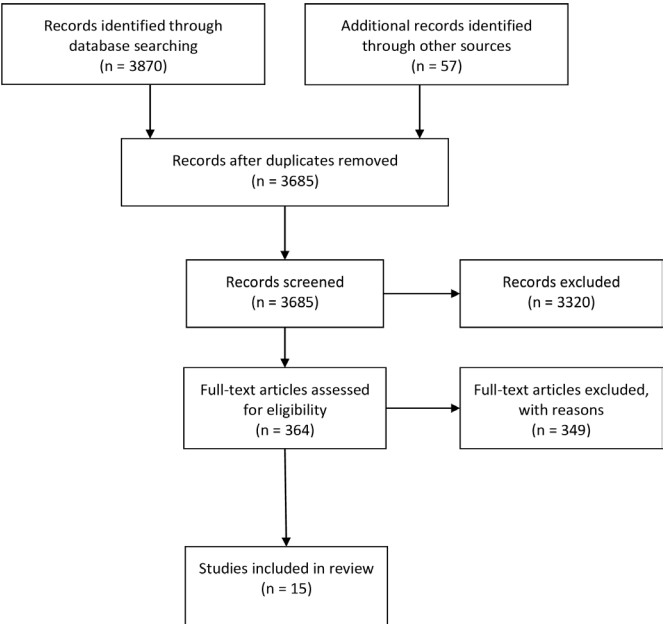

**Figure 1** Preferred Reporting Items for Systematic Reviews and Meta-Analyses flow diagram of literature review.

### Study selection and data extraction process
The data extraction process had two stages. Two authors (LS, JB) independently screened all titles and abstracts identified from searches to determine which met the inclusion criteria. They also independently screened full-text articles for inclusion or exclusion, with discrepancies resolved by discussion and by consulting a third author (PLS). All potentially relevant papers excluded from the review at this stage were listed as excluded studies with reasons. A data extraction form was developed to extract and appraise each study according to the aims of the study.

### Assessment of risk of bias of individual studies
Risk of bias was assessed using the Cochrane Collaboration tool[13] and was rated as low, unclear or high. Ratings of low risk of bias indicate high confidence in study findings. Ratings of unclear risk of bias indicate the presence of probable bias or flaws that raise reservations about study findings. High risk of bias are indicators of poor study quality, and results are considered with caution. One author (LS) assessed the risk of bias of the included studies which was reviewed by both additional authors (JB, PLS). Any disagreements were resolved by discussion to reach consensus.

### Synthesis of results
Thematic analysis was conducted within and across categories to identify key themes. All authors deliberated and agreed on these themes, as well as their validity and implications. Since one objective of this review is to inform policy, descriptive information about all interventions is presented and discussed in the results section. However, only those presenting low or unclear risk of bias are summarised and discussed in terms of their effectiveness.

Due to the heterogeneity of reported outcomes and risk of bias assessed in individual studies, a meta-analysis was not feasible.

Four themes were identified: (1) types of interventions, (2) composition of teams, (3) intervention effectiveness and (4) types of outcomes. Within intervention types, three categories of care types were identified: transitional care, case-management services and hospital at home. It is important to notice that these categories were not mutually exclusive and that some degree of overlap of the interventions' characteristics existed among the categories. These connections are highlighted in the results section.

### Patient and public involvement
Patients and the public were not involved in the design or analysis of this review.

### Systematic review reporting items
This systematic review follows Preferred Reporting Items for Systematic Reviews and Meta-Analyses's checklist, detailed in online supplementary appendix 3.

## RESULTS
### Study selection
A total of 3927 articles were identified. After screening titles and abstracts, 364 full-texts were reviewed, of which 15 articles met the inclusion criteria (figure 1).

### Study characteristics
Regarding types of interventions, the systematic review identified 15 interventions which were catalogued into three different care types: transitional care (n=8), case-management services (n=4) and interventions focusing on hospital-at-home (n=3).

Five studies were RCTs, eight were quasi-experiments (four prospective controlled pre–post design and one retrospective controlled pre–post design) and three were retrospective non-controlled observational studies. Four interventions took place in the USA,[14–17] while two interventions were set in the UK,[18 19] New Zealand[20 21] and Singapore.[22 23] Additionally, there was one intervention each in France,[24] Australia,[25] Denmark,[26] Mexico[27] and Hong Kong.[28] See table 1 for a summary of study characteristics.

### Risk of bias
Two RCT studies were considered as presenting low risk of bias,[20 26] while the others presented an unclear risk of bias due selective reporting that was discordant with the protocol,[25] due to implementation problems and power analysis[18] and due to the randomisation process.[15] Two quasi-experimental studies presented low risk of bias,[17 21] while the others were assessed as showing unclear risks due to a lack of randomisation[15 24 27] and programme implementation issues.[21] The remaining studies presented a high risk of bias due to elements of the methodological design or analysis[16 19 22 23 28] and

**Table 1** Summary of studies

| Study | Year | Methods | Number of participants | Demographics | Classification of intervention | Intervention | Team | Outcomes | Quality and bias assessment |
|---|---|---|---|---|---|---|---|---|---|
| Caplan et al[25] | 2004 | RCT | 739 | >75 | Case-management | Comprehensive geriatric assessment and multidisciplinary intervention on older patients sent home from the emergency department (ED). A nurse formulates a care plan, initiates urgent interventions and referrals, and presents the patient's history at a weekly interdisciplinary team meeting at which further interventions or referrals can be ordered. | Meetings attended by a geriatrician or a geriatric registrar, nurses, physiotherapists and occupational therapists. | All hospital admissions within 30 days Elective and emergency admissions (days to first emergency admission) Nursing home admissions Mortality Physical function (Barthel Index) Instrumental activities of daily living Cognitive function | Unclear risk of bias |
| Courtney et al[15] | 2009 | RCT and economic study (in additional paper) | 128 | >65 | Transition care | Comprehensive nursing and physiotherapy assessment and individualised programme of exercise strategies. Home visit by a nurse. Telephone follow-up commencing in the hospital and continuing for 24 weeks after discharge. | Nurse and physiotherapist. | Hospital readmissions and emergency General Practitioner visits | Unclear quality and risk of bias |
| de Stampa el al[24] | 2014 | Prospective controlled pre–post design | 428 | >64 | Case-management | Coordination of Professional Carefor the Elderly (COPA). Multidisciplinary comprehensive geriatric needs assessment. This includes an individual care plan, care management programmes, evidence-based protocols and regular reassessments of patient needs. | Case managers (geriatric nurses), primary care physician and a psychologist. | Unplanned hospitalisation (via ED) Any hospitalisation (1 year) Instrumental activity of daily living Activity of daily living Hierarchical cognitive performance Depression rating Aggressive behaviour Pain scale | Unclear risk of bias |
| Espinel-Bermudez et al[27] | 2011 | prospective controlled pre–post design | 70 | >60 | Hospital at home | Programme of domiciliary attention for chronically ill patients. Two educational sessions with patient carers. Home visits, provision of medical equipment and medication. Telephone line available for careers. | Two teams. The first is based in the hospital and second makes home visits (doctor, nurse, psychologist and a medical resident). | Hospital readmissions over 1 year (type, number of times, days to readmission and length of stay) Quality of life (sickness impact profile) | High risk of bias |
| Harris et al[20] | 2005 | RCT | 285 | >55 | Hospital at home | Seven days per week/10 hours per day nursing availability. 24 hours on-call geriatrician. Patient-centred planning. Daily nursing review and adjustment of individual care plan. Intensive home support with up to 24 hours support from a live-in home care professional. Multidisciplinary team support. Rehabilitation in the patient's home. A discharge handover to ongoing support services. | Nursing-led. Occupational therapist, physiotherapist and social workers. | Readmissions to hospital (first readmission in days 1–10,11–30 and 31–90) Physical and mental function (functional independence) Instrumental activities of daily living Mental status Self-reported recovery Health status (SF-36) Falls Satisfaction with intervention Costs | Low risk of bias |

Continued

**Table 1** Continued

| Study | Year | Methods | Number of participants | Demographics | Classification of intervention | Intervention | Team | Outcomes | Quality and bias assessment |
|---|---|---|---|---|---|---|---|---|---|
| Hendrix et al[16] | 2013 | Prospective controlled pre-post design | 47 | >60 | Transition care | Hospital visit before the patient's discharge to initiate a relationship with the patient and begin planning for home care. Home visits by team. Educational materials for caregivers and information on community-based resources. | Nurse practitioner, social worker, occupational therapist. | ED visits and rehospitalisation at 30 and 60 days after hospital discharge | Low quality and high risk of bias |
| Leung et al[28] | 2015 | Matched-control quasi-experimental study | 78 | Treatment arm: 80.2±7.7 Control arm: 80.5±8.5 | Hospital at home | The virtual ward physician might conduct a physician home visit within the first week after the patient was discharged. Based on the health assessment results, the virtual ward nurses planned the schedule of home visits which could range from daily to once every 2 weeks. | Nurses and physicians. | Unplanned emergency hospital readmissions, quality of life, emergency attendances | Low quality and unclear risk of bias |
| Low et al[23] | 2015 | Prospective controlled pre-post design | 259 | >65 | Transition care | An individualised patient-centred care plan drawn up for each patient after an initial home visit assessment. The frequency and timing of future home visits and the need to involve other members of the multidisciplinary team is identified during the initial assessment and communicated to all team members. Nurses make telephone call reviews based on patients' care needs and are accessible to patients by phone during office hours. | Multidisciplinary team comprising of a family physician, a nurse case manager, a physiotherapist, an occupational therapist, a speech therapist and a medical social worker. Nurses function as case managers. | ED attendances (after 3 and 6 months) Admissions to all hospitals (after 3 and 6 months) Length of stay (after 3 and 6 months) Specialist outpatient clinic visits (after 3 and 6 months) | Low quality and high risk of bias |
| Meret-Hanke[14] | 2011 | Prospective controlled pre-post design | 6992 | >65 | Case-management | Program of All-Inclusive Care for the Elderly (PACE): case-management services. Services provided in Day Health Centre (ie, recreational therapy nursing; social services, including caregiver training, support groups, respite care services and Dentistry). Attendance to centres varies from rarely to 7 days a week, depending on their care plans. | Comprehensive range of healthcare professionals. | Hospital use (as average days per month in hospital) (each 6 months for up to 2 years) | High risk of bias |
| Ong et al[22] | 2017 | Retrospective non-controlled observational | 107 | >51.9 | Case-management | Home-based medication reviews | Pharmacist and care coordinator. | Hospital admissions ED visits Length of stay (days) | High risk of bias |
| Robinson et al[21] | 2015 | Retrospective controlled pre-post design | 5239 | >65 | Transition care | Transition intervention: postdischarge component (telephone assessment, education, support by team) to identify problems not dealt with during discharge, assist with self-management, and ensure appropriate health and social support. | A geriatrician, a pharmacist and cultural support workers. | 28-day readmission rates ED visits | Low risk of bias |
| Rytter el al[26] | 2010 | RCT | 331 | >78 | Transition care | Home visit by a GP and a district nurse 1 week after discharge, followed by two contacts after three and 8 weeks. | GP and nurse. | Readmission rate within 26 weeks after discharge Control of medication, evaluated 12 weeks after discharge | Low risk of bias |

Continued

**Table 1** Continued

| Study | Year | Methods | Number of participants | Demographics | Classification of intervention | Intervention | Team | Outcomes | Quality and bias assessment |
|-------|------|---------|------------------------|--------------|--------------------------------|--------------|------|----------|------------------------------|
| Sahota et al[18] | 2017 | RCT and economic study | 250 | >70 | Transition care | Community In-Reach rehabilitation and Care Transition. A comprehensive assessment of the patient's ability to perform different tasks and creation of a rehabilitation plan. Form links with the appropriate community services to ensure a smooth and effective discharge. Following discharge, team home visits to assess the level of rehabilitation required. Further follow-up visits as deemed necessary and appropriate referral to additional community services. | A senior occupational therapist (transition coach), a senior physiotherapist and an assistant practitioner, linked directly to a social services practitioner, working across multiple boundaries with patients and their carers. | Hospital length of stay Readmission (28 and 91 days postdischarge) Total time in bed days Functional ability Comorbidity and health-related quality of life Costs | Unclear risk of bias |
| Stranges et al[17] | 2015 | Retrospective cohort study | 1144 | >60 | Transition care | Phone call 2 to 4 days after discharge. The patient seen in clinic, ideally within 1 week of discharge, by a social worker and a health professional. Home visit by health professional, who conducts a modified geriatric assessment with a focus on the reason for hospital admission. | Geriatric physicians, nurse practitioners, clinical pharmacists and social workers | 30-day readmission rates | Low risk of bias |
| Young et al[19] | 2005 | Prospective controlled pre–post design | 1648 | >63 | Transition care | An intermediate care (IC) service. Joint care management team (multi-disciplinary, multi-agency) assesses need and purchases services for individuals delivered through a primary care trust. Patients accepted for IC are then assessed by each discipline in the team and a care plan is developed to be delivered by the care assistants. Patients receive input for up to 6 weeks, according to need. | Nurses (5), occupational therapists (2), care assistants (15), a dietitian (half time) and access to psychiatric nurses. | Nottingham Extended Activities of Daily Living score Barthel Index Hospital Anxiety and Depression score Mortality (3, 6 and 12 months) Readmission to hospital (3, 6 and 12 months) Readmission to hospital number of days (3, 6 and 12 months) New institutional care placements (3, 6 and 12 months) | Unclear risk of bias |

baseline differences.[14 27] The assessment of risk of bias of individual studies is detailed in online supplementary appendix 4.

## Types of interventions
### Transitional care

Transitional care is defined as a set of actions to ensure the coordination and continuity of healthcare as patients transfer between different locations; from hospital to nursing homes or between different levels of care within the same location.[29] Eight studies looked at transitional care interventions as a primary focus. Low et al[23] present a hospital-based intervention where patients received a comprehensive needs assessment in the home, followed by an individualised care plan that included medical and nursing care, patient education and coordination of care with other clinical specialists and community services. Nursing care included visits and follow-up phone calls. Robinson et al[21] describe an intervention that included an improved discharge process with two components: the identification of high risk patients and a postdischarge component that consisted of a nurse telephone assessment, along with education, and support by community nurses on the first and third days postdischarge. Sahota et al[18] present a programme with a predischarge functionality assessment tool and a prospective rehabilitation plan. After discharge, the service team performed home visits to assess progress and to provide appropriate referral to additional community services. Young et al[19] report on a city-wide intermediate care service where a joint care management team assessed patient needs and purchased support and rehabilitation from specialised teams.

Courtney et al[15] present an intervention portraying a comprehensive patient assessment and individualised care plan developed in consultation with the patient, health professionals, family and caregivers. The intervention also had a weekly follow-up telephone call after discharge. Hendrix et al[16] present an intervention led by a nurse practitioner providing postdischarge medical care to patients for a time-limited period, with the addition of an occupational therapist and a social worker, who conducted separate home visits. Rytter et al[26] describe a joint home visit involving a GP and district nurse conducted 1 week after discharge and two contacts with the GP in the third and eighth weeks after discharge. Finally, Stranges et al[17] describe a multidisciplinary team of geriatric physicians, nurse practitioners, clinical pharmacists and a social worker who assisted patients transitioning to the community. The intervention began with a pharmacist's phone call 2 to 4 days after discharge, followed by a clinic appointment with a social worker and geriatrician or nurse within a week of discharge.

In summary, transitional care focused on the postdischarge component and covered a range of different actions such as the elaboration of individualised patient-centred care plans,[15 23] telephone assessment and education[21] and home visits for short periods of time.[17–19 23] These could lead to referrals to other services such as rehabilitation

or additional community services.[18 19] Elements of transitional care are also apparent in some of the following intervention types.

### Case-management services

Case management is a targeted, community-based, proactive approach to care that involves case-finding, assessment, care planning and care co-ordination.[30] Three studies had case-management services as their focus. Caplan et al[25] provide information on a comprehensive geriatric assessment performed in an emergency department, in addition to follow-up visits by a hospital-based multidisciplinary outreach team over 4 weeks. Meret-Hanke[14] describes comprehensive medical and social services to targeted community-dwelling older individuals. The programme organised their services in a Day Health Centre which members attended with varying frequency (from occasionally to daily), depending on their care plans. De Stampa et al[24] examine a programme consisting of a physician and a case manager who conducted a home-based comprehensive geriatric assessment and produced individualised care plans using evidence-based interdisciplinary protocols. Finally, Ong et al present a home-based medication review programme led by a pharmacist, accompanied by a care coordinator, in liaison with other medical specialists. The programme provided pharmacological counselling, and the identification of drug-related problems.[22]

All case-management interventions shared some characteristics: they included geriatric assessments, the elaboration of individualised care plans and coordination between care managers and/or multidisciplinary meetings where further interventions or referrals could be made.[24 25] While transitional care was not a focal point of this intervention type, the function of case management (as defined) was clearly concerned with ensuring a smooth transition between different services using various tools.

### Hospital at home interventions

Hospital at home encompasses the active treatment at home by care professionals of people who would otherwise be admitted to hospital, and early supported discharge from hospital.[24] Three studies present information regarding hospital at home interventions. Espinel-Bermúdez et al[27] describe a programme with visits by a multidisciplinary team, the provision of educational sessions and a telephone line available to carers, in addition to the provision of medical equipment and medication to the patients. Harris et al[20] describe an intervention that supported 10 hours per day of nursing, 24 hours on-call geriatricians, daily patient-centred planning by nurses and, if needed, the upgrading of individual care plans to offer intensive home support with up to 24 hours of a live-in home care professional. Finally, Leung et al[28] present a virtual ward service, whereby a nurse and a GP separately visit the home in the first week after discharge. The nurse visited the home every 2 weeks, providing

psychosocial support to both patients and their carers. Additionally, a hotline consultation service was put in place. There are clear links to transitional care within this category.

## Composition of teams

A wide variety of multidisciplinary teams, in terms of composition and size, was observed across the studies. Two interventions were pharmacist-led, accompanied by other staff such as care coordinators[22] or cultural workers in a context of diversity.[21] All other studies described nurses playing leading roles, such as case managers.[20 23 24] Geriatricians were present in only one study[25] while family or primary care doctors were part of the intervention in three studies.[23 24 27] Other health workers such as physiotherapists, occupational therapists, psychologists, speech therapists and dietitians were found in all studies.[14 18 19 23–25 27] Only three interventions included social workers.[14 18 20]

## Intervention effectiveness
### *Effects on hospital admissions and readmissions*

Only the results of three studies assessed as presenting low risk of bias are presented. Only one intervention categorised as transitional care[17] found a significant reduction in 30-day readmission rates among matched compliers (n=217, 11.7% vs 17.3%, treatment and control respectively; p<0.001) and also a positive delay in terms of time to readmission, which was significantly longer among those receiving the intervention (18±9 days compared with 12±9 days with usual care; p=0.015). Contrarily, neither Harris *et al*[20] (n=208, 76.8% vs 69.2%, treatment and control respectively; p>0.05) nor Robinson *et al*[21] (n=2486, regression discontinuity parameter estimation=−1.60 (SD 3.10); p>0.05) found statistically significant differences in terms of readmissions.

### *Healthcare expenditure*

Harris *et al* report the costs of the hospital at home (NZ\$6524) intervention were almost double standard hospital care (NZ\$3525),[20] while Sahota *et al* found a net monetary benefit of £1932 (95% CI £2134 to £5863), although this has to be interpreted with caution given the wide confidence intervals.[18] Additionally, Courtey *et al*'s intervention estimated an average cost saving of \$333 for 24 weeks' period per patient, and a net-monetary-benefit per individual of \$7907.[31]

### Type of outcomes

Ten studies reported on different lengths of time of hospital readmissions,[16–21 26–28] while six reported hospital admissions.[14 18 22–25] Only one study presented both outcomes.[18] Six studies reported emergency department admissions[15 21–25] and five measured different lengths of hospital stay.[14 18 22 23 27] Additionally, three studies measured emergency GP visits[15 16 28] and one also measured control of medication.[26]

Other outcomes, such as functional status and quality of life or depression, were measured in several prospective studies.[18–20 24 27 28] Mortality was assessed in two studies,[19 25]

and there was a costs analysis in three studies[15 18 20]; one of them in a separate study.[31] Finally, patient satisfaction was assessed in one study.[20]

Results were assessed over a wide range of time periods. One study started measuring results 10 days after the intervention started,[20] while the most common follow-up period ranged from 1 to 6 months.[14–21 23 25 26 28] Two studies extended the follow-up period to 1 year[19 27] and one measured the effects after 2 years.[14]

## DISCUSSION
### Hypothesis traps limit relevant results

Different concurrent explanations may explain the limited number of studies that produce significant statistical differences. One plausible explanation, known as the 'omitted variable bias', notes that outcome measures such as hospital readmissions and length of stay are moderated by other confounding variables that are usually not captured in large-sample studies. Conversely, the same outcomes cannot be easily observed in smaller-sample in-depth studies, due to the impact of sample size in hypothesis testing.

Another explanation is the 'outcome selection trap', which refers to the practice of identifying relevant and easy-to-collect outcomes (therefore generating less measurement error) in intervention studies. However, those outcomes could be considered excessively distant from the nature of the interventions. For instance, valuable low-impact interventions such as follow-up phone calls should not be assessed in terms of reduction of hospitalisation, but in terms of other subjective measures such as quality of life or process-related outputs such as correct use of medication.

A third explanation is the 'causality gap', which refers to the inability of many of those interventions to establish a clear theoretical causal connection between, on one hand, activities and resources and, on the other, outputs and outcomes. This includes an inability to estimate the proportion of resources, time and activities needed to achieve specific measurable changes, such improvements in health status.

A fourth explanation is the 'timeframe gap', whereby studies do not analyse effects over a sufficiently long period to observe changes that accrue over lengthier time frames.

A fifth explanation is the 'instrument gap' which refers to the limitations of chiefly using outcomes such as hospital admissions, readmissions or length of stay to measure interventions and not developing adequate tools to identify and measure frailty or including the older person's perspective.

### The diversity and limitations of interventions

The review included different interventions focusing on reducing the avoidable displacement from home of older people, varying from a range of complex designs such as hospital at home programmes to simpler ones

such as medication reviews. These interventions occurred over different lengths of time, including short-term transitional care services and longer ones such as hospital at home assistance and case-management interventions. The number of caregivers, the permanent, periodical or sporadic nature of their work, and their specialisations varied significantly.

All 15 studied interventions were based almost entirely within the health system and were provided by a wide range of health specialists. With the exception of three studies,[16–18] there was no integration between mainstream health services and other agencies, such as housing services or social care.

At the same time, there was an absence of preventive interventions. All the quasi-experiments and RCT review interventions viewed the hospital setting as the 'initial status' of the intervention. Only one study emphasised preventive care through regular clinical monitoring.[25] This is especially relevant for interventions that focus on a wide range of older people with different health statuses and different degrees of functional dependency.

With respect to the intervention types (transitional care, case management and hospital at home), the overlaps were unsurprising, given their definitions and the complex arrangement of services and actions needed to care for frail older people within an integrated care context. Some parallels with other broad-based initiatives such as intermediate care can be seen. In the UK where intermediate care concepts have been operationalised, they include four types of intervention: crisis response, home-based care, bed-based care and reablement.[32] The category of home-based care in this review corresponds with the hospital at home category.

### Assessing interventions with the 'avoidable displacement from home' framework

The avoidable displacement from home framework appears to be a useful lens through which to assess integrated care interventions. That said, the three key values of avoidable displacement from home (person-centredness, place-based care and sustainability) did not emerge prominently in this review. None of the interventions reviewed referred to all of these elements. Two management service interventions[24 25] might be characterised by a person-centred approach with a focus on individual rehabilitation and care plans. However, neither study specifically assessed these interventions from a person-centred perspective. This was an important omission which might have been averted through more robust application of the avoidable displacement from home framework. Likewise, including place-based care as an analytical theme would have revealed degrees of service integration and fragmentation, especially for interventions that seek to integrate external care with services at home.[32] It was not possible to determine the degree of service integration for most interventions, but the available evidence for the medication review intervention indicated that it did not fully satisfy Kodner and Kyriacou's definition of integrated

care.[7] In terms of intervention sustainability, many studies focused on short-term interventions and outcomes (1 to 6 months) which was not necessarily compatible with meeting the chronic, complex needs associated with older age. In terms of economic sustainability, only three studies provided any element of economic analysis,[18 20 31] reflecting the numerous challenges of quantifying cost benefits for integrated care interventions.[33]

### Comparison with previous literature reviews

Four previous systematic reviews[34–37] examined the effectiveness of different types of interventions to prevent hospitalisations based on home care and demonstrated the limitations of current impact evidence, in terms of consistent positive outcomes. Huntley et al[38] reviewed evidence about alternatives to acute care, such as paramedic/emergency department-based interventions and community hospitals. They reported that impact evidence was limited by a lack of systematic data on outcomes and costs for patients, health professionals or carers. In a review of integrated care interventions for older people, Baxter et al found inconsistent effects on admissions, length of stay and costs.[39]

### Limitations of this review

This review looks at interventions to reduce avoidable displacement from home that are conducted in home settings. It does not consider other settings, including those that may be applied in hospitals and nursing homes, which should be the focus of future reviews.

All the articles but one in this review studied healthcare systems in high-income countries, even though low or middle-income countries contain a larger number of older people with health and care needs. This may reflect the limited number of languages included in the study, although that is unlikely to have excluded a large number of studies.

Empirical evidence in this area is predominately from quasi-experiments and observational studies, while studies assessed as having a low risk of bias were an exception.

Although the search strategy focused on hospital avoidance, all included interventions were related to admissions and readmissions. Other search strategies could inform on interventions promoting hospital admission avoidance.

This review is limited to quantitative findings due to the heterogeneity of the qualitative studies in terms of frame of reference and focus of enquiry. Future reviews should seek to address this challenge in order to maximise the available evidence.

### CONCLUSIONS AND IMPLICATIONS

One purpose of this review was to establish the potential validity of the concept of avoidable displacement from home. The review was not able to identify robust impact evidence either in terms of quantity or quality from the studies presented. Therefore, the evidence cannot be

considered sufficiently strong to inform policy or interventions seeking to reduce avoidable displacement from home. The paucity of evidence does not result from the limited number of potentially relevant interventions. Rather, it reflects the complexity of these interventions and a lack of systematic data collection. As such, this review identifies an urgent need for systematic monitoring and data management, as well as enhanced data collection. Data should be derived from a more careful selection of measurement instruments, implementation strategies and robust methods for evaluating multifaceted interventions in complex populations.

**Acknowledgements** The authors thank the invaluable contribution of the reviewers: Gemma Hughes, Asangaedem Akpan and Ken Hillman as well of BMJ Open editor.

**Contributors** All authors made a relevant contribution during the process. LS: lead systematic reviewer conducting all stages of the review and was responsible for the initial draft of protocol and draft of paper. JB: second reviewer; contributing to discussion as the review progressed; commenting and editing on the drafts of the paper. PLS: secured funding, idea of current work, third reviewer and contributing to discussion as the review progressed; commenting and editing on the drafts of the paper. All authors reviewed the submission to address editors and peer reviewers comments.

**Funding** This paper presents research funded by Eastern ARC Consortium and Medical Research Council (grant reference number MR/R024219/1).

**Disclaimer** The views expressed are those of the authors and not necessarily those of the Eastern ARC Consortium and Medical Research Council

**Competing interests** None declared.

**Patient consent for publication** Not required.

**Provenance and peer review** Not commissioned; externally peer reviewed.

**Data availability statement** All data relevant to the study are included in the article or uploaded as supplementary information.

**ORCID iD**
Lucas Sempé http://orcid.org/0000-0002-0978-6455

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
