## [Reviewer comments · BMJ Open]

ARTICLE DETAILS

TITLE (PROVISIONAL)	Multidisciplinary interventions for reducing the avoidable displacement from home of frail older people: a systematic review
AUTHORS	Sempé, Lucas; Billings, Jenny; Lloyd-Sherlock, Peter

VERSION 1 – REVIEW

REVIEWER	Gemma Hughes University of Oxford, UK
REVIEW RETURNED	13-Apr-2019

GENERAL COMMENTS	General comments: 1. This paper employs the framework of avoidable displacement from home (ADH) to assess the effectiveness of multi-disciplinary interventions for older people with frailty or multi-morbidity. The framework of ADH appears to have the potential to be a very useful way of assessing integrated care interventions. As this seems to be a novel and helpful concept, I would like to see more detail about the methodology used to develop this framework (or reference to a source of this information) in the body of the paper, and a related reference in the abstract. I would also suggest a brief rationale for use of this novel framework rather than the more commonly used outcome measures of (avoidable) hospital admissions.2. I think a more detailed rationale for the search strategy used to identify reviews should be provided to clarify the following points:a) How were reviews relating to multi-disciplinary teams (MDTs) identified? Given the emphasis on MDT in the title of the paper I would have expected MDT to have been included in the search teams. The authors state in the abstract that interventions without multi-disciplinary teams were excluded, however it's not clear how such interventions were initially included.b) You include only quantitative studies (and state on page 3 that evidence is pre-dominantly from quasi-experiments and observational studies). I think it would be helpful to clarify i) why you have included only quantitative studies and ii) why you have chosen not to undertake a meta-analysis. The heterogeneity of reported outcomes is an implicit reason not to undertake meta-analysis but I think this should be explicitly stated.c) How is ADH incorporated into your search terms and if it is not can you explain how you use hospital admissions as a proxy for ADH (if indeed that is what you have done).3. I would also like to see a clearer explanation of the way in which you identified the different themes and in particular a stronger justification for distinguishing between the types of intervention as
--

you do (the four types given are transitional care, case management, hospital-at-home, and medication review). There seems to be some degree of overlap between these types (for example care management is provided as part of transitional care in Young et al., 2005). I also think there needs to be further justification for including medication review as a category of its own when it also appears to have included care coordination.

Line by line comments follow:

Page 2 (Abstract)

Line 6 I suggest that ADH (avoidable displacement from home) is written in full rather than abbreviated and that the novel nature of this framework is noted here.

Line 45 Strengths – Please clarify what you mean by the ‘new policy framework’ – do you mean ADH? It is not clear to me in what way this is a ‘policy’ framework. However I do agree that the use of ADH is one of the strengths of this review and should be explicitly mentioned here.

Page 3

Introduction

I think (line 23) you need to join the dots in this paragraph between the outcome of avoided hospitalisations and the interventions to integrate care – you need to state that the assumption is that these will result in hospitalisations being avoided.

Line 38 – I think you need to add here a brief outline of how ADH has been developed or reference the source for this

Page 4

Line 3 Clarify here how ADH was defined in the search

Line 18 Eligibility criteria – what is your rationale for only including quantitative studies?

Line 25 Outcomes included are hospital-based – what makes ADH different from avoided hospital admissions?

Line 48 Search strategy – how did you identify multi-disciplinary interventions without using this as a search team?

Page 5

Line 34 I think you need to justify why you chose to conduct a thematic analysis and not a meta-analysis (I think this is entirely justifiable but need a line to explain).

Page 13

Line 44 Transitional care – this section is much longer and has a great deal more operational detail than the other kinds of intervention. Is all of this detail necessary? As noted above I think that you need to justify your categories including explaining any overlap between them, noting the interacting and potentially not mutually exclusive nature of these ‘themes’. You might also want to distinguish between the categories you have identified and those of a) intermediate care and b) admission avoidance schemes.

Page 15

Line 48

Again I think it is worth clarifying that there is likely overlap between these categories. Although this is the only study that primarily focused on medication review at home, I would have expected that the comprehensive geriatric needs assessment conducted as part of case management (de Stampa et al., 2014) for example, would have included a medication review. Likewise for a hospital at home scheme (e.g. Espinel-Bermudez et al 2011) which includes provision of medication – was a medication review included?

	Page 17 Line 6 I would suggest you need to say more about what a multi-disciplinary team is and how you identified these in the studies. Also to clarify the relationship between the MDT and the different types of intervention as MDTs are also considered as an intervention in and of themselves. For example, are the MDTs prerequisites for the provision of the different interventions you have identified? Line 22 You present the statistical findings of the studies without doing any meta-analysis, therefore I would suggest a tighter narrative analysis and more interpretation (e.g. which kinds of interventions seem to be most effective?). Page 18 Line 35 I would like to see a clearer use of the ADH framework. For example, which studies (if any) measured person-centredness/place-based care/sustainability and what were the findings? Line 46 You introduce the concept of fragmentation here – although this isn't part of the ADH framework. Are you suggesting fragmentation is the antithesis of care at home? Page 19 Line 6 Comparison with previous literature reviews: It would be helpful to comment on the findings of this review as compared with:  1. Huntley AL, Thomas R, Mann M, et al. Is case management effective in reducing the risk of unplanned hospital admissions for older people? A systematic review and meta-analysis. Family Practice. 2013. 2. Pearson M, Hunt H, Cooper C, Shepperd S, Pawson R, Anderson R. Providing effective and preferred care closer to home: a realist review of intermediate care. Health & Social Care In The Community. 2015;23(6):pp577-593. 3. Baxter S, Johnson M, Chambers D, Sutton A, Goyder E, Booth A. The effects of integrated care: a systematic review of UK and international evidence. BMC health services research. 2018;18(1):350. Typographical errors appear to occur in the following locations: Page 2 Line 19 Page 3 line 5 Page 16 line 23
--	--

REVIEWER	Asangaedem Akpan Institute of Ageing & Chronic Disease University of Liverpool England
REVIEW RETURNED	30-Apr-2019

GENERAL COMMENTS	A very interesting review that introduces a different perspective on assessing interventions to maintain older people with frailty at home. The use of a narrative synthesis to explore the quantitative type studies done for themes was in my opinion a very good approach There were some minor typos and grammatical errors which am sure the authors can easily correct. May be useful to have someone unconnected with the review to read through as that will be a fresh pair of eyes and more likely to spoy these minor typos and grammatical errors.
---

	Frailty and integrated care as concepts may benefit from elaborating on their definition and citing some references. For example frailty can be determined by a range of frailty scales/scores and it is not clear in this review if that was looked for. There are over 50 definitions if not a lot more of what integrated care is and again this has relevance for making broad statements as to what works and does not. In defining their ADH framework it may have been useful to have considered health outcomes that matter to older people. In summary a very good review and I would recommend it for publication
--	--

REVIEWER	Ken Hillman University of New South Wales Australia
REVIEW RETURNED	03-May-2019

GENERAL COMMENTS	This is an important and original study. It emphasises the paucity of information available at the moment about this issue inferring that a lot more work both in development, implementation and evaluation needs to be carried out. Perhaps a short sentence could be added emphasising this, especially the need for the parallel development and implementation or the need for policy, implementation strategies and robust evaluation. A minor point - use the full name of ACD in the abstract as it is not necessarily a widely used term. There are many reasons to criticise outcomes such as hospital admissions, length of stay and mortality. Perhaps a short sentence along the lines of the need to identifying those patients who are nearing the end of life using existing tools, especially those including frailty. The outcomes would be different for these patients especially if the tools could be used as a flag for honest and empathetic discussions and construction advance care directives around their attitudes and beliefs. Armed with this information many elderly people would prefer to be supported in their homes. Increasingly this is becoming important for patients and carers and needs different outcome measures as they do not necessarily want to be admitted to hospital and mortality becomes expected and a normal outcome necessitating whether the death met the expectations of the person. I realise due to space constraints, all of this cannot be added However the strength of this article is the emphasis on the lack of current research and this article could add some suggestions about future directions including the importance of what we hope to achieve by these interventions, including the recognition of patients near the end of life. The acceptability of alternatives depends largely on issues such as funding and whether they meet people's expectations. This is variable across countries and there are other models such as in Germany, Japan, China and Karala in India which may not have robust evaluation as yet but perhaps a brief sentence about other national models which may be examined more closely in the future.
---

VERSION 1 – AUTHOR RESPONSE

Reviewer 1	
I would like to see more detail about the methodology used to develop this framework (or reference to a source of this information) in the body of the paper, and a related reference in the abstract	This has been done. In the framework sub-section, we have clarified that this review is linked to a wider process of concept development. A related conceptual paper is currently under review with a different journal.
I would also suggest a brief rationale for use of this novel framework rather than the more commonly used outcome measures of (avoidable) hospital admissions.	See above.
How were reviews relating to multi-disciplinary teams (MDTs) identified? Given the emphasis on MDT in the title of the paper I would have expected MDT to have been included in the search teams. The authors state in the abstract that interventions without multi-disciplinary teams were excluded, however it's not clear how such interventions were initially included.	This has been specified in the sub-section on eligibility criteria.
How is ADH incorporated into your search terms and if it is not can you explain how you use hospital admissions as a proxy for ADH (if indeed that is what you have done).	Since ADH is a new concept, it was not specifically included as a search term. The different specific elements that make up ADH were included.
I would also like to see a clearer explanation of the way in which you identified the different themes and in particular a stronger justification for distinguishing between the types of intervention as you do (the four types given are transitional care, case management, hospital-at-home, and medication review). There seems to be some degree of overlap between these types (for example care management is provided as part of transitional care in Young et al., 2005).	We have developed definitions to provide a clear explanation of the categories used and reduced the number of categories to three as recommended. We also address the issue of overlaps between categories.
I also think there needs to be further justification for including medication review as a category of its own when it also appears to have included care coordination.	We agree that the medication review study is better included as part of the case management category. This change has been made.
Abstract Line 6 I suggest that ADH (avoidable displacement from home) is written in full rather than abbreviated and that the novel nature of this framework is noted here.	This has been done
Abstract Line 45 Strengths – Please clarify what you mean by the 'new policy framework' – do you mean ADH?	This has been done
P3 Introduction Line 23 I think you need to join the dots in this paragraph between the outcome of avoided hospitalisations and the interventions to integrate care – you need to state that the assumption is that these will result in hospitalisations being avoided.	This has been done. Below it is possible to see the correction: Given the wide diversity of integrated care interventions and their complex interactions with other parts of health and social care systems, it is

	important to establish clear parameters about where they are situated in relation to different care settings and to clarify outcomes of interest. (...) More broadly, an avoidable displacement from home can be understood as the consequence of failures to deliver proper care that permits older people to remain in their homes for as long as possible when this is in their best interest. It represents a comprehensive approach to address health and social care as part of a single, integrated system.
P3 Introduction Line 38 – I think you need to add here a brief outline of how ADH has been developed or reference the source for this	This has been done (see above)
P4 Line 3 Clarify here how ADH was defined in the search	This has been done
P4 Line 18 Eligibility criteria – what is your rationale for only including quantitative studies?	In our initial search we did not exclude qualitative studies. The review team includes a member who primarily works with qualitative methods. However, the team elected to focus exclusively on studies with quantitative findings due to the extreme heterogeneity of the qualitative studies in terms of frames of reference and focus of enquiry. Including qualitative studies would now require us to go back to a much earlier stage of the process and produce a very different paper. We have now added an acknowledgment of this limitation to our study design and we hope this is satisfactory.
P4 Line 48 Search strategy – how did you identify multi-disciplinary interventions without using this as a search team?	This has been explained, as follows: A multidisciplinary team was defined as a formal team of two or more people from different professions working together to deliver their service.
P5 Line 34 I think you need to justify why you chose to conduct a thematic analysis and not a meta-analysis (I think this is entirely justifiable but need a line to explain).	This has been done, as follows: Due to the heterogeneity of reported outcomes and risk of bias assessed in individual studies, a meta-analysis was not feasible.
P13 Line 44 Transitional care – this section is much longer and has a great deal more operational detail than the other kinds of intervention. Is all of this detail necessary?	The section was shortened. Comparing the number of studies in each section, there is a proportionality on the three categories.
As noted above I think that you need to justify your categories including explaining any overlap between them, noting the interacting and potentially not mutually exclusive nature of these 'themes'. You might also want to distinguish between the categories you have identified and those of a) intermediate care and b) admission avoidance schemes.	This has been done by establishing definitions at the beginning of each section. We were not able to identify mainly admission avoidance schemes. That is one of the findings of the review regarding the lack of preventive care.
P15 Line 48 Again I think it is worth clarifying that there is likely overlap between these categories. Although this is the only study that primarily focused on medication review at	This has been done as follows: With respect to the intervention types (transitional care, case management and hospital at home), the overlaps were unsurprising, given their

home, I would have expected that the comprehensive geriatric needs assessment conducted as part of case management (de Stampa et al., 2014) for example, would have included a medication review. Likewise for a hospital at home scheme (e.g. Espinel-Bermudez et al 2011) which includes provision of medication – was a medication review included?	definitions and the complex arrangement of services and actions needed to care for frail older people within an integrated care context. Some parallels with other broad-based such as intermediate care can be seen. In the UK where intermediate care concepts have been operationalised, they include four types of intervention: crisis response, home-based care, bed-based care and reablement [32]. The category of home-based care in this review corresponds with the hospital at home category.
P17 Line 6 I would suggest you need to say more about what a multi-disciplinary team is and how you identified these in the studies. Also to clarify the relationship between the MDT and the different types of intervention as MDTs are also considered as an intervention in and of themselves. For example, are the MDTs prerequisites for the provision of the different interventions you have identified?	MDT is not considered a type of intervention but a condition of inclusion in our review. MDT works as a proxy of integration of services. That has been clarified in the text (as explained before)
P17 Line 22 You present the statistical findings of the studies without doing any meta-analysis, therefore I would suggest a tighter narrative analysis and more interpretation (e.g. which kinds of interventions seem to be most effective?).	It has been done. The section was reduced only to focus on three studies with an assessment of having a low risk of bias. Only the results of three studies assessed as presenting low risk of bias are presented. Only one intervention categorised as transitional care [17] found a significant reduction in 30 day readmission rates among matched compliers (n = 217, 11.7% vs 17.3%, treatment and control respectively; p <.001) and also a positive delay in terms of time to readmission, which was significantly longer among those receiving the intervention (18 ± 9 days compared with 12 ± 9 days with usual care; p = .015). Contrarily, neither Harris et al [20] (n=208, 76.8% vs 69.2%, treatment and control respectively; p >.05) nor Robinson et al [21] (n=2486, Regression discontinuity parameter estimation= -1.60 [SD 3.10]; p >.05) found statistically significant differences in terms of readmissions.
P18 Line 35 I would like to see a clearer use of the ADH framework. For example, which studies (if any) measured person-centredness/place-based care/sustainability and what were the findings?	This has been addressed The avoidable displacement from home framework appears to be a useful lens through which to assess integrated care interventions. That said, the three key values of ADH (person-centredness, place-based care and sustainability) did not emerge prominently in this review. None of the interventions reviewed referred to all of these elements. Two management service

	interventions [24,25] might be characterised by a person-centred approach with a focus on individual rehabilitation and care plans. However, neither study specifically assessed these interventions from a person-centred perspective. This was an important omission which might have been averted through more robust application of the ADH framework. Likewise, including place-based care as an analytical theme would have revealed degrees of service fragmentation, especially for interventions that seek to integrate external care with services at home [32]. It was not possible to determine the degree of service integration for most interventions, but the available evidence for the medication review intervention indicated that it did not fully satisfy Kodner and Kyriacou's definition of integrated care [8]. In terms of intervention sustainability, many studies focused on short-term interventions and outcomes (1 to 6 months), which was not necessarily compatible with meeting the chronic, complex needs associated with older age. In terms of economic sustainability, only three studies provided any element of economic analysis [18,20,31], reflecting the numerous challenges of quantifying cost benefits for integrated care interventions [33].
P18 Line 46 You introduce the concept of fragmentation here – although this isn't part of the ADH framework. Are you suggesting fragmentation is the antithesis of care at home?	Fragmentation is understood as the opposite of place-based service integration. It has been clarified in the text. In the text: Likewise, including place-based care as an analytical theme would have revealed degrees of service fragmentation, especially for interventions that seek to integrate external care with services at home [32]. It was not possible to determine the degree of service integration for most interventions, but the available evidence for the medication review intervention indicated that it did not fully satisfy Kodner and Kyriacou's definition of integrated care [8].
P19 Line 6 Comparison with previous literature reviews: It would be helpful to comment on the findings of this review as compared with: 1. Huntley AL, Thomas R, Mann M, et al. Is case management effective in reducing the risk of unplanned hospital admissions for older people? A systematic review and meta-analysis. Family Practice. 2013. 2. Pearson M, Hunt H, Cooper C, Shepperd S, Pawson R, Anderson R. Providing effective and preferred care closer to home: a realist review of intermediate care. Health & Social	This has been done, as follows: Four previous systematic reviews [34–37] examined the effectiveness of different types of interventions to prevent hospitalisations based on home care and demonstrated the limitations of current impact evidence, in terms of consistent positive outcomes. Huntley et al [38] reviewed evidence about alternatives to acute care, such as paramedic/emergency department-based interventions and community hospitals. They reported that impact evidence was limited by a lack of systematic data on outcomes and costs

Care In The Community. 2015;23(6):pp577-593. 3. Baxter S, Johnson M, Chambers D, Sutton A, Goyder E, Booth A. The effects of integrated care: a systematic review of UK and international evidence. BMC health services research. 2018;18(1):350	for patients, health professionals or carers. In a review of integrated care interventions for older people, Baxter et al found inconsistent effects on admissions, length of stay and costs [39].
Reviewer 2	
Frailty and integrated care as concepts may benefit from elaborating on their definition and citing some references. For example frailty can be determined by a range of frailty scales/scores and it is not clear in this review if that was looked for.	This has been done, as follows: Later life is associated with an increased risk of frailty, poor health and functional limitation[2]. In this paper, frailty refers to a distinctive health state related to the ageing process in which multiple body systems gradually lose their in-built reserves [3]. There are several definitions of integrated care: this paper follows Kodner and Kyriacou's [7] who define it as a "... set of techniques and organisational models designed to create connectivity, alignment and co-ordination within and between the cure and care sectors at the funding, administrative and/or provider levels.... for patients with complex problems".
In defining their ADH framework it may have been useful to have considered health outcomes that matter to older people.	The ADH framework includes other relevant health and social outcomes to older people (e.g. person-centredness leads to QoL). However, in this specific study we focused on the effects, which are measured in terms of avoiding hospitalisation.
Reviewer 3	
It emphasises the paucity of information available at the moment about this issue inferring that a lot more work both in development, implementation and evaluation needs to be carried out. Perhaps a short sentence could be added emphasising this, especially the need for the parallel development and implementation or the need for policy, implementation strategies and robust evaluation	This has been done as follows: The paucity of evidence does not result from the limited number of potentially relevant interventions. Rather, it reflects the complexity of these interventions and a lack of systematic data collection.
There are many reasons to criticise outcomes such as hospital admissions, length of stay and mortality. Perhaps a short sentence along the lines of the need to identifying those patients who are nearing the end of life using existing tools, especially those including frailty. . The outcomes would be different for these patients especially if the tools could be used as a flag for honest and empathetic discussions and construction advance care directives around their attitudes and beliefs. Armed with this information many elderly people would prefer to be supported in their homes. Increasingly this is becoming important for patients and carers and needs different outcome measures as they do not necessarily want to be admitted to	This has been commented on in the paper as follows As such, this review identifies an urgent need for systematic monitoring and data management, as well as enhanced data collection. Data should be derived from a more careful selection of measurement instruments, implementation strategies and robust methods for evaluating multifaceted interventions in complex populations.

hospital and mortality becomes expected and a normal outcome necessitating whether the death met the expectations of the person. I realise due to space constraints, all of this cannot be added However the strength of this article is the emphasis on the lack of current research and this article could add some suggestions about future directions including the importance of what we hope to achieve by these interventions, including the recognition of patients near the end of life.	
--	--

VERSION 2 – REVIEW

REVIEWER	Gemma Hughes University of Oxford, UK
REVIEW RETURNED	02-Sep-2019

GENERAL COMMENTS	I think the avoidable displacement from home framework is very useful and look forward to seeing other publications from these authors.
---

REVIEWER	Asangaedem Akpan Institute of Ageing and Chronic Disease University of Liverpool England
REVIEW RETURNED	17-Jul-2019

GENERAL COMMENTS	My comments have been addressed as well as the other reviewer comments from what I have read
--